# HIV Replication Under High-Level Cabotegravir Is Associated with the Appearance of 3′-PPT Mutations, Circular DNA Transcription and Recombination

**DOI:** 10.3390/v16121874

**Published:** 2024-11-30

**Authors:** Xierong Wei, Jonathan T. Lipscomb, Ariana Santos Tino, Mian-er Cong, Susan Ruone, Meghan L. Bentz, Mili Sheth, Gerardo Garcia-Lerma, Jeffrey A. Johnson

**Affiliations:** 1Laboratory Branch, Division of HIV Prevention, National Center for HIV, Viral Hepatitis, STD and TB Prevention, Centers for Disease Control and Prevention, Atlanta, GA 30329, USA; wax3@cdc.gov (X.W.); eyk1@cdc.gov (J.T.L.); mgq1@cdc.gov (A.S.T.); mianer.cong@gmail.com (M.-e.C.); fyh1@cdc.gov (S.R.); jng5@cdc.gov (G.G.-L.); 2Anyar Inc., 2113 Lewis Turner Blvd, Fort Walton Beach, FL 32547, USA; 3Advanced Diagnostics and Biotechnologies Branch, Division of Core Laboratory Services and Response, National Center for Emerging and Zoonotic Infectious Diseases, Centers for Disease Control and Prevention, Atlanta, GA 30329, USA; npk7@cdc.gov (M.L.B.); vgg1@cdc.gov (M.S.)

**Keywords:** cabotegravir resistance, CAB, 3′-PPT, LTR circles, HIV recombination

## Abstract

The HIV integrase inhibitor, dolutegravir (DTG), in the absence of eliciting integrase (int) resistance, has been reported to select mutations in the virus 3′-polypurine tract (3′-PPT) adjacent to the 3′-LTR U3. An analog of DTG, cabotegravir (CAB), has a high genetic barrier to drug resistance and is used in formulations for treatment and long-acting pre-exposure prophylaxis. We examined whether mutations observed for DTG would emerge in vitro with CAB. HIV-1IIIB was cultured in paired experiments of continuous high (300 nM) CAB initiated 2 h or 24 h after infection. After eight months of CAB treatment, no int resistance was detected. Conversely, HIV RNA 3′-PPT mutants were detected within one month and were the majority virus by day 98. The appearance of 3′-PPT variants coincided with a rapid accumulation of HIV 1-LTR and 2-LTR circles. RNA amplification from the 3′-LTR TAR identified transcripts crossing 2-LTR circle junctions, which incorporated the adjacent U5 sequence identical to the 3′-PPT mutants. 3′-PPT variants were only identified in LTR circles and transcripts. Additionally, we found evidence of linear HIV and LTR circle recombination with human DNA at motifs homologous to 3′-PPT sequences. HIV persistence under CAB was associated with transcription and recombination of LTR circle sequences.

## 1. Introduction

Integrase strand transfer inhibitors (INSTIs) are a class of antiretroviral drugs (ARVs) intended to inhibit human immunodeficiency virus (HIV) integrase (IN) activity, thereby preventing virus integration into host DNA [1]. The integrated proviral DNA serves as the template for subsequent virus production and is, therefore, a crucial step in establishing infection. Multiple units of IN are required to catalyze viral DNA to yield reactive 3′-OH ends and direct the strand transfer at each end of the virus genome. For HIV, the number of IN units involved can vary from four to more and together form the intasome [2,3,4]. Regardless of the number of units involved, all intasomes have a reactive core to catalyze integration, initiating the cleavage of DNA. Once DNA is cleaved, the core becomes exposed, allowing access by INSTIs where the INSTIs can stack against the catalytic domains to compete with target DNA transfer [5].

Earlier-generation INSTIs, raltegravir (RAL) and elvitegravir (EVG) are known to generate various mutations in IN associated with two general pathways to resistance, involving codons Q148/G140 or N155/E92, and for RAL also Y143/E92. To counter resistance observed with RAL and EVG use, second-generation INSTIs, bictegravir (BIC), dolutegravir (DTG), and cabotegravir (CAB), were developed that are capable of tightly stacked, stable interaction with the IN-DNA catalytic space that creates a high genetic barrier to resistance. CAB, which has a greater dissociation half-life than DTG, is formulated for oral or injectable administration and was recently FDA-approved as a component of a bi-monthly two-drug regimen for the maintenance of HIV suppression [6]. CAB is also approved as long-acting pre-exposure prophylaxis administered in bi-monthly injections for the prevention of HIV infection [7].

The high barrier to int resistance with second-generation INSTIs has led to observations of unusual occurrences of resistance mutations in vitro with DTG outside of int, in the guanosine hexamer (6-G) at the 3′ end of the 3′-polypurine tract (3′-PPT), that were recently evaluated by Malet, et al. [8]. Subsequently, similar mutations were observed in an individual experiencing virologic failure (VF) with DTG maintenance monotherapy [9]. Here, we examined the more stable analog of DTG, CAB, for evidence of int drug resistance mutations under a continuous high therapeutic-relevant concentration in culture. We also examined for sequences changes in the HIV-1 3′-PPT and the IN-binding termini of the virus long terminal repeats (LTR). We elaborate on the involvement of episomal HIV DNA that has also been raised by others on the maintenance of drug-resistant variants under CAB treatment.

## 2. Materials and Methods

### 2.1. Virus Cultures Under Cabotegravir Treatment

To examine for drug resistance selection with INSTI dosing, CEMx174 cells (cultured in RPMI 1640 medium supplemented with 10% fetal bovine serum) were infected with wildtype HIV-1IIIB (5.0 × 10^8^ HIV cp/10^6^ cells, cells and virus were obtained from the NIH AIDS Reagent Program through BEI resources). For both CAB and RAL, treatments were initially designed with escalating doses for each inhibitor. However, the virus did not survive with this CAB culture; therefore, we repeated treatment initiating with 300 nM CAB, a concentration that is ~315 times the in vitro EC90 (CAB EC90 = 0.95 nM/EC50 = 0.56 nM) [10] and is in the range for the protein-adjusted EC90 (median = 411 nM) [11]. Two cultures were separately continuously treated with CAB, at 2 h or 24 h after inoculation with virus (cultures 1 and 2, respectively), to ascertain any differences in outcomes in which HIV integration had already occurred (culture 2) compared to blocking initial integration with early treatment (culture 1). Virus-induced cytopathic effect (CPE) was observed at week 2 of infection, and thereafter, media with drug was refreshed weekly. For comparison, a set of virus cultures was treated with RAL both to demonstrate that the culture conditions would generate classic int resistance mutations and to assess if RAL elicited any changes in the 3′-PPT. RAL treatment began at 0.3 nM (passage 1) and was doubled with each passage up to 614.4 nM (passage 12). Increasing passages of CAB were also performed as a preamble to the study reported here.

### 2.2. HIV Genome Amplification

We analyzed for CAB-selected mutations in int gene, the IN-binding regions in the HIV long terminal repeats (LTR), the 3′-PPT, and sections crossing the circular LTR junctions [12]. Nucleic acids were extracted from culture supernatants using the QIAamp Viral RNA Mini Kit, and from cells using the QIAamp DNA Blood Mini Kit (QIAGEN, Inc.). For viral RNA amplification, nucleic acid extracts were first DNase I treated using the DNA-free™ DNA Removal Kit (Ambion, Austin, TX, USA) for 1 h at 37 °C. PCR and nested PCR of the post-digest DNA were performed to ensure the completeness of DNase digestion. For RT-PCR amplification, cDNA synthesis was performed at 42 °C for 1 h using MuLV Reverse Transcriptase (Life Technologies, Carlsbad, CA, USA), followed by PCR amplification using the GeneAmp High Fidelity mix (Applied Biosystems) with annealing at 54 °C for 30 s and extensions for 2 min at 72 °C. Due to an interruption in sourcing, the later time points used Superscript IV One-step RT-PCR mix (Invitrogen, Waltham, MA, USA) with reverse transcription at 50 °C for 10 min. LTR 5′R-U5 and 3′U3-R cDNA, proviral nef-3′U3LTR, and 2-LTR DNA amplification were performed for 40 cycles. Primer sequences in 5′-3′ orientation are provided in Table 1 and mapped in Figure 1 for reference.

### 2.3. Sequencing of HIV Integrase Gene and LTRs

Int, nef, LTR, and circle junction amplifications from viral DNA and RNA were sequence analyzed. For bulk Sanger sequencing, amplified products underwent chain termination labeling using BigDye v1.1 (Applied Biosystems, Foster City, CA, USA) and were resolved on the ABI Prism 3130XL. For deep sequencing, the Illumina Nextera XT DNA Library Preparation Kit was used and resolved on the MiSeq. For the integrated DNA and circle recombinant sequencing, libraries were generated with SMRTbell Express Template Prep Kit 3.0 and sequenced on the PacBio Sequell II using Sequell II Binding kit 3.2. All sequences were analyzed in CLC Genomics Workbench with the Low Frequency Variant Detection tool (QIAGEN, Redwood City, CA, USA) set at a 2% cutoff.

### 2.4. Strategies to Detect Integrated HIV

Due to overwhelming representation of LTR circles in the DNA preparations, several exclusion steps were necessary to minimize primer competition from episomal HIV sequences to better identify integration events. DNA was digested overnight at 37 °C then at 50 °C for 4 h with BamHI, BclI, ClaI, and SalI endonucleases to fragment LTR circles to smaller sizes while preserving the genomic regions in LTR-gag and nef-LTR amplified by their respective primers. The remaining regions of circular DNA were cleaved into fragments of <4 kbp for size for exclusion using a diluted AMPure PB beads-based size selection method at a 3.1× ratio. Sized DNA fractions were amplified for sequencing using both HIV gag and nef biotinylated primers paired with either human ALU primers or a primer directed against a blunt end-ligated unique linker (NEXTFLEX^®^ Rapid DNASeq Kit 2.0 kit, Revvity, Waltham, MA, USA) to detect DNA fragments containing HIV sequences. Amplicons containing the biotinylated HIV-specific amplicons were column purified (Miltenyi µMACS) prior to performing the sequencing reactions.

### 2.5. In Vitro Culture Fitness Competition in the Absence of CAB

To assess the replicative capacity of wildtype HIV-1IIIB in the absence of CAB as compared to the variant with the complete AGCAGT mutation in the 3′-PPT, described below, dilutions of mutant:wildtype mixtures at a ratio of 90% mutant:10% wildtype were seeded in duplicate onto CEMx174 cells. Nucleic acid extracts from culture supernatants were DNase I treated, and the frequency of 3′-PPT mutant sequences in expressed vRNA was assessed by deep sequencing at days 0, 5, 7, and 10 of culture. The growth comparison of HIV with the fully mutated 3′-PPT against wildtype virus over time was described elsewhere [13,14] and the relative amount of expressed virus was used in calculating the replicative capacity of the mutant using online tools (https://www-binf.bio.uu.nl/rdb/fitp.html and https://indra.mullins.microbiol.washington.edu/cgi-bin/vgrc.cgi, accessed on 1 August 2024).

## 3. Results

### 3.1. Assessment of In Vitro Int Resistance with RAL and CAB Treatment

We examined increasing RAL dosing over time to ensure our culture system could successfully select for HIV int resistance. With RAL int resistance emerged to bulk sequence detectable frequencies beginning with the Q184R mutation at passage 9 (day 127, [RAL] = 76.8 nM). At passage 10 (day 138, [RAL] = 153.6 nM), both Q148R and E138EK were detected (Figure 2), and at passages 11 (day 147) and 12 (day 155), the S230R mutation also emerged together with E138K and Q148R ([RAL] = 307.2 and 614.4 nM, respectively). We did not find evidence of RAL eliciting changes in the 3′-PPT by bulk sequencing. While the RAL culture was selected for classic INSTI mutations as doses increased, the escalating CAB treatment caused a loss of virus replication after three passages. Therefore, for CAB treatment, we switched to initiating with a high dose equivalent to what is experienced with HIV treatment. Sanger and NGS (Illumina MiSeq) sequence analysis of the HIV-1IIIB int gene from each culture passage under CAB treatment identified no resistance mutations over the course of eight months. Cell cultures under RAL dosing were terminated at passage 12, with viral CPE still evident.

### 3.2. Appearance of 3′-PPT Mutations In Vitro Under Steady-State Therapeutic Levels of CAB

#### 3.2.1. HIV RNA

For CAB culture 2 in which the drug was added 24 h after virus inoculation, the sequence of the RT-PCR-2 (see Table 1) product, which amplifies from both circular and linear template, identified low-level 3′-PPT mutations in vRNA by deep sequencing at day 14 of culture (Figure 3). At this time point, the mutations in the 6-nucleotide motif were present at a frequency of 8% (culture VL = 3 × 10^7^ cp/mL) (see Table 2). Under sustained CAB treatment, mutations slowly accumulated to a frequency of 26% between days 28 and 35 of culture. By day 98, 62% of culture 2 vRNA had the mutated 3′-PPT sequence, AGCAGT, which peaked at 78% on day 105 (VL = 3 × 10^9^ cp/mL). After this time point, the mutation frequency began to dwindle slowly in that it was observed at 53% and 43% on days 120 and 127, respectively. We noted, however, that only 10% of these 3′-PPT hexamer sequences were complete and in-frame, largely due to frameshift deletions at the first A in the tract. For culture 1, in which CAB was added 2 h after virus inoculation, the RT-PCR-2 sequences reproducibly divulged a different mutation pattern in the 3′-PPT, instead of the AGCAGT mutation observed for culture 2, culture 1 nearly exclusively generated the mutated PPT sequence, GGAGCA. Additionally, unlike the high prevalence of frameshift deletions identified from culture 2, for culture 1, there were very few deletions in the tract, which resulted in intact GGAGCA peaking at 94% of 3′-PPT sequences at day 120. Only 3% of the culture 1 sequence had the AGCAGT sequence at this later time point.

Because the RT-PCR-2 can amplify viral RNA that originated from either integrated provirus or LTR circles, it was necessary to identify if, specifically, LTR circles were contributing to the expressed viral RNA pool. Focusing first on culture 1, for which integration would not have been initiated due to adding CAB at two hours, transcripts from nef to the 2-LTR circle junction were identified in 14% of the sequences however, complete AGCAGT mutations were at only 4% and this rapidly decreased to 0.3% and lower. We found that the transcripts from 2-LTR circles in culture 2 primarily were deleted in the segment that spanned from immediately 3′ of the PPT to the junction. Of the sequences that were intact, 83% had the complete AGCAGT 3′-PPT mutation at day 98. In contrast, the GGAGCA 3′-PPT mutation was seen in 13% of 2-LTR-specifc sequences, and all were intact. Additionally, evidence of 2-LTR transcripts was detected by RT-PCR-3b, which originated downstream 3′ to the 3′-LTR TAR and amplified to the end of the 5′-LTR U3. These sequences revealed vRNAs incorporating the 3′-U5 through the circle junction (GTAC) to the end of 5′-U3. Four weeks after removing CAB from culture transcripts with the 3′-PPT mutation disappeared.

#### 3.2.2. HIV DNA

PCR-2 performed on DNA from both cultures first identified 3′-PPT mutations at passage 2 (day 14) at a frequency of 14% (Figure 3). At day 98, 42% and 18% of sequences for cultures 1 and 2, respectively, had the intact AGCAGT mutation, and at day 127 the frequencies dropped marginally to 22% and 12%. For culture 1, it was expected that all viral DNA existed as unintegrated episomes given that CAB treatment was initiated two hours after virus inoculation. Viral DNA with deletions were also observed throughout the nef-LTR region, some exceeded 100 bp and concentrated around the 3′-PPT. A notable finding was that a 4 bp fragment of the 3′ end of U5 was also present contiguous with mutated 3′PPT. As with RNA, this primer pair cannot discern whether the amplified DNA was from provirus or circular episomes; therefore, we compared the proportion of mutations from this reaction to the proportion detected in reactions specific for circular HIV.

In performing cross-junction PCR-3a, we detected a rapid accumulation of 1- and 2-LTR circles under 300 nM CAB. The majority of the nef-to-gag amplicons from circular DNA had wildtype LTR sequences flanking the circle junction (represented in Figure 3 and Figure 4). Also present were circles with tandem repeats in U3 that 3′-flanked the junction. From the 2-LTR circle-specific PCR-7 product, culture 2 had 10–15 times the frequency of intact 2-LTR circles with the AGCAGT mutation than did culture 1. As was observed for RNA, these sequences had a significant drop-off in reads at the end of PPT, with much lower representation of sequences that continued to the circle junction. Culture 1, while having fewer 2-LTR circles than identified for culture 2, had a 2–4 times greater proportion of complete 3′-PPT mutations in general, which again were almost entirely comprised of the GGAGCA motif. From culture 2 the pan-nef RT-PCR-2 reaction identified a lower frequency of the intact AGCAGT variant than was identified in circular DNA, which likely reflected contributions of the WT sequence from a virus that had integrated at the beginning of culture. Again, for culture 2, we did not identify the GGAGCA mutation sequence that was predominant in the 3′-PPT from culture 1.

Treatment with 300 nM CAB was ended at day 127, until which time HIV expression was consistently high (10^9^ copies/mL) in both cultures. We saw that for both cultures, mutant 3′-PPT HIV RNA expression waned over four weeks after cessation of CAB treatment. The cross-junction transcripts were no longer detected one month after cessation of CAB treatment, and two weeks later, HIV LTR circle DNA disappeared.

PCR amplification of digested DNA size selected for ~4–20 kbp fragments (Figure A1) that were randomly tagged with a common linker primer identified HIV sequences that ended precisely at the 2-LTR junction, providing evidence that HIV 2-LTR circles also existed as cleaved linear forms. One PCR product from the long sequence analyses identified a sequence from the nef primer that incorporated a complete single LTR that proceeded into gag to nucleotide position 1272, at which point the sequence transitioned to a second nef domain. It was found that at that transition, there is a seven nucleotide stretch (TAGAAGA) that is identical in gag and nef. Together with the repeated sequences in the 3′-U3 mentioned above, this provides additional evidence of LTR circle recombination at sites of sequence homology. Other fragments that include integrations with human DNA are described below.

### 3.3. Evidence of the Origin of 3′-PPT Mutations

In examining the hexameric mutations that appeared in the 3′-PPT from culture 2, the nucleotide sequence could not be explained by mutation due to the natural random error rate of reverse transcriptase. Upon closer examination of both the mutated 3′-PPT sequence and the region surrounding the 2-LTR junction, we recognized that the mutated 3′-PPT matched a sequence of six nucleotides 5′-adjacent to the circle junction (Figure 3) [15,16,17]. The involvement of 2-LTR circles in generating 3′-PPT mutations with DTG treatment was recently reported as a result of reverse transcription product extension by Dekker et al. [18]. Of note, the palindrome centered at the 2-LTR junction, CAGT|ACTG (Figure 3), is associated with high-efficiency endonucleolytic cleavage in vitro. The resultant AGCAGT mutation in the 3′-PPT introduces the same high-efficiency IN cleavage sequence two nucleotides before the start of the 3′-U3 (…AGCAGT|ACTGG…). Thus, two cleavage sites were now present, one each at the circle junction and at the 3′-PPT.

The GGAGCA 3′-PPT mutation that was predominant in culture 1 is a partial composition of the culture 2 mutation except that the last two nucleotides (GT) of the 3′ U5 are missing. We note, however, that the four-nucleotide change (AGCA) matches the next four upstream nucleotides in 3′-U5 that abut the junction of 2-LTR circles. Aberrant processing of the 3′ terminus of the HIV genome by IN may have resulted in the loss of the dinucleotide, which is typically preserved as an overhang after cleavage.

### 3.4. Recombination of 2-LTR Circles with Human DNA

PCR amplifications of gag-nef segments identified insertions of human sequences from culture 2 in which human DNA was juxtaposed with HIV precisely at the junction site of 2-LTR circles. Notably, unlike the sequence preference reported for IN-directed integration [19], the termini of the human sequences that were inserted at the 2-LTR circle junctions were highly homologous to the HIV 3′-PPT that flanks the U3. There was also a sequence in which human DNA, mapped to chromosome 13, connected to HIV in nef and continued past the 2-LTR junction to the second distal U3 region. The positioning of the nef sequence is reminiscent of a 20-bp nef insert adjacent to the U3 sequence of an HIV integrant that was also identified (see below). Human ALU-HIV integration-specific PCRs were also performed to examine the composition of sequences surrounding integrated HIV loci and identified HIV 5′ termini insertions at 10+ distinct human loci (partially represented in Figure 4), which mapped to chromosomes 1, 9, 12, 13, 19, and X (SeqID NG_013262.1). Of note, similar to what was observed with insertions into circular HIV DNA, several sequences directly adjacent to the integrated HIV 5′-LTR were comprised of a stretch of five or six G’s preceded by four A’s. This polypurine region is similar to the sequence and positioning of the wildtype 3′-PPT adjacent to the viral DNA HIV 3′-LTR. In Sequence 2 of Figure 4, we see the mutated 3′-PPT along with the adjacent 3′U5 in the 2-LTR product. Additionally, we identified one integrant containing 38 nucleotides of contiguous nef sequence upstream of the homologous PPT region and another integrant that contained 20 bp of the beginning of nef inserted between U3 and the upstream human sequence. There was also evidence of traditional integration site sequences adjacent to the 5′ and 3′ ends of integrated HIV, which did not show homology to the region around the PPT. We could only identify a few integrated proviruses 3′-end reads that extended to the 3′-PPT, and these revealed the wildtype poly-G sequence. This may represent the wildtype virus, which was integrated prior to adding CAB. We did not identify ALU PCR products supporting integration into human sequences with culture 1.

### 3.5. Fitness of the CAB-Selected 3′-PPT Mutant Virus

Conservation of the 3′-PPT motif is critical for efficient HIV second-strand DNA synthesis [7,8,20,21]. For this reason and to better understand if the mutated 3′-PPT variants that emerged have the propensity to persist in an environment absent of integrase inhibitor, we examined the replicative capacity of virus containing the predominant 3′-PPT mutant containing the 6-G motif replacement, AGCAGT, as compared to wildtype virus in tissue culture competition experiments. Duplicate experiments initiated with 90% mutant/10% wildtype virus cultured in the absence of CAB yielded very similar results of a rapid decline in mutant vRNA such that the 3′-PPT variant comprised only ~8–11% of the total viral population at day 5 of culture (Figure 5). By day 10 of culture, wildtype virus represented more than 93% of the viral RNA. Due to the plateauing of the mutant virus decline after day 5, data from the first 5 days of culture were used in calculating the relative fitness of the mutant. During that period, the wildtype virus expanded 4.4-fold while the mutant/wildtype ratio declined 84-fold. Inputting the relative ratios and VLs into the fitness calculation website (https://www-binf.bio.uu.nl/rdb/fitp.html, accessed on 26 October 2024) yielded a selection coefficient (s) of −1.792 and a relative fitness (1 + s) of −0.792. Hence, replication of the variant with the substituted 3′-PPT is highly inefficient in the absence of 300 nM CAB. Moreover, when we analyzed both vRNA and p24 expression of the mutant virus at 300 nM CAB culture passage 12 (day 105), we observed that the ratio of p24/vRNA was 120 times lower for mutant virus under CAB treatment than for wildtype HIV-1IIIB propagated in the absence of antiretrovirals.

## 4. Discussion

With CAB used in key antiretroviral interventions such as a component of a suppressive dual ARV regimen comprising a long-acting pre-exposure prophylaxis against HIV acquisition, and being under consideration for other extended-use formulations, examining the potential for HIV to circumvent sustained CAB dosing is needed. In this system with high viral expression, the effectiveness of a therapeutic CAB concentration in blocking HIV integration was exhibited by the rapid accumulation of 1- and 2-LTR circles. The CAB treatment initiated 24 h after virus inoculation, more mimicking an established infection, compared to initiating at 2 h was to examine for an impact of integrated virus on resistance emergence. Here, we further demonstrate in vitro the high genetic barrier of CAB to int resistance relative to earlier-generation INSTIs in that after eight months of continuous CAB treatment, no mutations were identified in int. In contrast, RAL was selected for typical INSTI mutations by 18 weeks of culture. We elaborate on both 1-LTR and 2-LTR circle involvement in HIV persistence in the presence of a sustained therapeutic-equivalent concentration of CAB.

We identified under CAB treatment evidence of rapid emergence of viral mutations at minority levels outside of int, in the 3′-PPT located adjacent to the 3′-LTR U3, which accumulated to high frequency after 3 months. For culture 1, when CAB was added at 2 h after virus inoculation, the sequence change identified in the 3′-PPT, GGAGCA, was very similar to that previously reported from a person who virologically failed under DTG monotherapy, with the difference being the frameshift of one 5′ nucleotide [8,9]. Mutations in this region have been reported to be associated with 1-LTR circle templates [22,23]. Surprisingly, when CAB treatment was initiated 24 h post-infection, we consistently identified a different 3′-PPT variant with an in-frame sequence of AGCAGT. Viruses with the complete AGCAGT 3′-PPT mutation exhibited inferior replicative capacity in the absence of CAB relative to wildtype virus, though under a CAB concentration inhibitory to wildtype virus, high VLs were achieved with the mutant. We did, however, observe reduced p24 protein expression relative to vRNA levels for the mutant virus, which may have been due to the mutated 3′-PPT sequence, which resides in nef. The mutations cause a change in amino acids for two codons, from glycine–glycine to the hydrophobic residues alanine-valine (G95A/G96V), possibly disrupting the function of nef protein to support both the production and cell surface accumulation of p24 for virus particle assembly [24]. When CAB treatment was stopped, the mutant viruses disappeared from culture by four weeks, thus exhibiting a classic resistance mutation detriment in the absence of drug pressure.

Our interpretation of the appearance of 3′-PPT mutations and their positioning relative to the 3′-LTR U3 IN cleavage site was that they were derived from sequences adjacent to U3 at the LTR circle junctions. The more compelling evidence was from the 2-LTR specific reaction in which the AGCAGT sequence replacing the six G’s in the 3′-PPT exactly matched the six nucleotides in U5 positioned 5′ to the canonical 2-LTR junction cleavage site (see Figure 3). Hence, the composition and positioning of the mutation relative to the U3 cleavage site suggested the replacement of the 3′-PPT hexamer by the junction-adjacent U5 sequence, possibly by recombination in trans with sequences that cross the circle junction and through homology with the 3′-U3. We also note that the human sequences integrated adjacent to 3′-PPT-homologous sequences were also comprised of extended polypurine-rich stretches, which are favored sequences for integration.

To examine for evidence that may support transcriptional recombination from LTR circle products, we specifically examined virus transcripts that crossed the 2-LTR circle junctions. We did identify RNA transcripts spanning from the 3′-LTR R to 5′-LTR U3 and, in doing so, copied the terminus of U5 that is identical to the 3′-PPT mutation observed from culture 2. Coinciding with the increase in the 3′-PPT variant, 2-LTR circle templates were largely missing the region beginning immediately downstream of the 3′-PPT to the circle junction. Excision of this region in 2-LTR circles involves two important sites for IN recognition and high-efficiency cleavage. The natural cleavage site at the 5′ of U3, which is also the requisite sequence for linearizing 2-LTR circles [25], and a second site in which this cleavage sequence is replicated in the PPT as a result of the sequence replacement. We do not know if there is an advantage in HIV having a second cleavage site at the 5′ end of the 3′-LTR. The products of the dual-cleaved 2-LTR sequence would resemble a 1-LTR template and a separated complete copy of the removed 3′-LTR. Whether the freed LTR or 1-LTR remnant can be scavenged for some function is unknown.

By blunt-end digestion and virus sequence to random linker PCR, we found evidence of linear forms of HIV DNA terminated at the junction, which supports that junction cleavage also occurs in the presence of CAB. The functional relevance of linearized 2-LTR circles is unknown; however, it may be possible that they can serve as non-integrated templates for transcription. Moreover, we identified human sequences inserted both into the junction site and in nef of 2-LTR circles, thereby providing evidence of 2-LTR circle recombination with human DNA. We also saw evidence of human ALU-HIV sequences that contained fragments of nef embedded between the human sequences and the HIV 5′-LTR.

The mechanism of 3′-PPT mutation formation postulated here would suppose that LTR circle DNA can serve as a template for virus recombination and transcription. The identification of a circle sequence with a second nef region inserted out of place in gag provides evidence that intermolecular recombination involving LTR circles occurred. Whether these circles also recombined with integrated HIV found in culture 2 is unknown. While the virus under CAB produced 3′-PPT variants with the complete mutant sequence, other mutations reflected the variability of sequence processing adjacent to the 2-LTR junction. Additional evidence of imprecise processing near the circle junction was observed with 3′-PPT variants that included pieces of adjacent nef sequences in later time points, which may imply variation in sequence processing by IN as unique target templates arise.

From the evidence obtained under sustained CAB treatment, we found: (1) for culture 1 HIV 1-LTR circles primarily and for culture 2 2-LTR circles predominantly coincided with the appearance of 3′-PPT mutations in viral RNA, (2) the HIV DNA circles accumulated 3′-PPT mutations over three to four months, (3) 2-LTR circle-specific RNA transcripts were detected that contained the 3′-PPT mutations and crossed the circle junction to incorporate the U5 domain identical to the 3′-PPT mutant, (4) the episomal HIV DNA existed both in closed circles and linearized forms, (5) recombinant sequences of human DNA were inserted at the junction site and in nef of 2-LTR circles, and (6) a few loci of HIV integrated into human DNA that appeared to have occurred at sites of homology between either linear 2-LTR and/or circular 1-LTR HIV and human sequences. HIV sequence processing by IN followed by homologous recombination with human DNA could be an outcome of CAB interfering with the strand transfer step that would normally occur with integration.

We propose the 3′-PPT motif was replaced by translocated LTR circle junction-adjacent sequences as a result of the recombination of LTR circle products in the presence of treatment-relevant concentrations of CAB. The sequence difference between the predominant mutant species in culture 1, GGAGCA, as compared to culture 2, AGCAGT, is the missing 3′ “GT” that is typically inserted between U5 and the junction cleavage location in 2-LTR circles, thereby keeping in culture 1 guanosines in the first two positions of the hexamer. This might imply variations in tRNA primer removal by RNase H or IN processing at the terminus of HIV prior to the formation of the episomal circles [12,26]. Of note, the hexamer in culture 1 is repeated at a few sites in the HIV genome, including in nef. This alternative sequence at the 3′-PPT, however, does not result in a duplication of the canonical palindrome required for high-efficiency cleavage by IN. The prevalence of this distinct PPT sequence in culture 1 may imply a predominant contribution by 1-LTR circles in transcript production when integration is initially prevented. However, the contribution of 2-LTR circles early in culture 1 cannot be excluded, given the evidence of those sequences in transcription and recombination.

There are limitations to our investigation. HIV LTR circles may be more stable in culture, allowing for mechanisms to occur that might have less opportunity in vivo. We do not know if there is a role for HIV integrated early in culture 2 in virus breakthrough. The frequency of 3′-PPT mutation was marginally lower when comparing total nef amplified products to that of LTR circle-specific nef sequences, suggesting wildtype integrated virus contributed to detectable sequences. We were only able to obtain very few long reads involving human DNA integration at HIV 3′ termini that extended upstream to the 3′-PPT. From these products, we identified only wildtype PPT sequence, and it is unknown if rare homologous integration events involving the mutated 3′-PPT may have happened; however, no concrete evidence of integration was identified from culture 1. We did not perform whole transcript analysis to fully ascertain what other molecular mechanisms could have led to the generation of infectious 3′-PPT variants in the two cultures. It was, in fact, difficult to amplify long transcripts, which may have been a result of variability in episomal sequence expression generating an overwhelming quantity of incomplete transcripts.

New generation INSTIs, with their high potency and barrier to resistance, hold great promise in helping to dampen the HIV epidemic. While the CAB-associated resistance we present here is from in vitro experimentation, it cannot be ignored that 3′-PPT mutations were also identified in a person experiencing virologic failure under maintenance monotherapy with its analog, DTG [9]. When reducing the number of antivirals in maintenance therapy, the capacity to suppress the virus may be more sensitive to variations in INSTI drug concentrations at anatomic sites [27,28]. These findings further demonstrate the tenacity of HIV in exploring avenues to circumvent antiretroviral inhibition and the prudence needed to investigate mechanisms that may lead to reduced susceptibility to highly potent antiretroviral drugs.

## Figures and Tables

**Figure 1 viruses-16-01874-f001:**
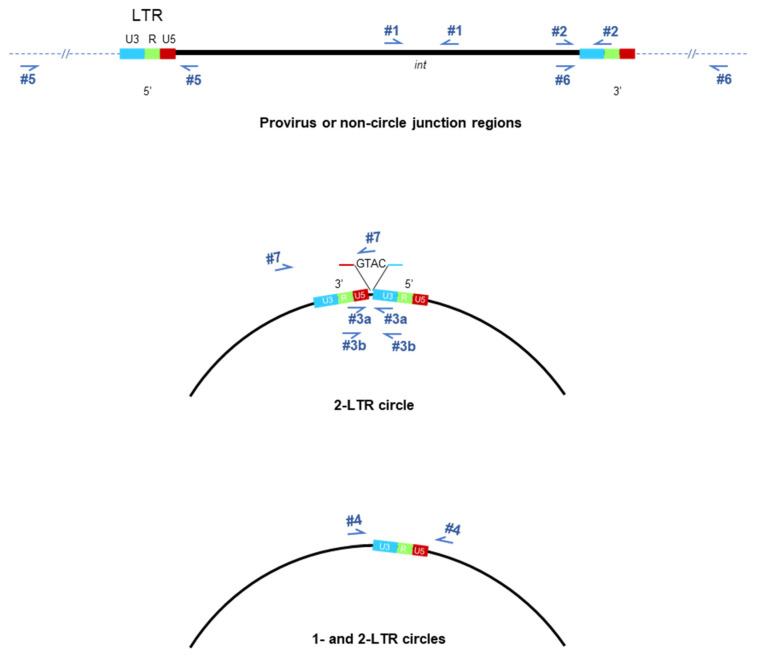
Primer map for oligonucleotides used for sequence amplification. The reaction numbers refer to the primer pairs listed in Table 1.

**Figure 2 viruses-16-01874-f002:**
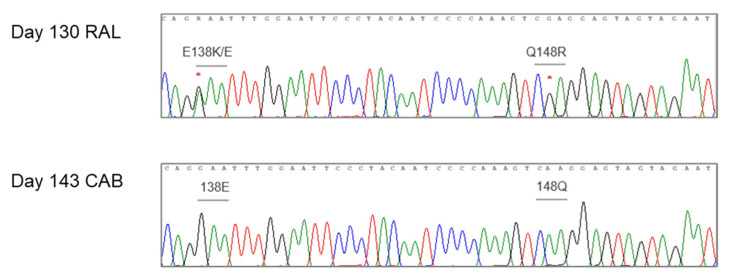
Alignment of consensus sequence of integrase (IN) region from HIV-1 grown in the presence of RAL or CAB. Known mutations (*) for IN resistance emerged with RAL treatment by culture passage 9, whereas no IN mutations were generated with CAB over one year in culture. Nucleotide peak colors: cytosine (blue), adenosine (green), thymidine (red), guanosine (black).

**Figure 3 viruses-16-01874-f003:**
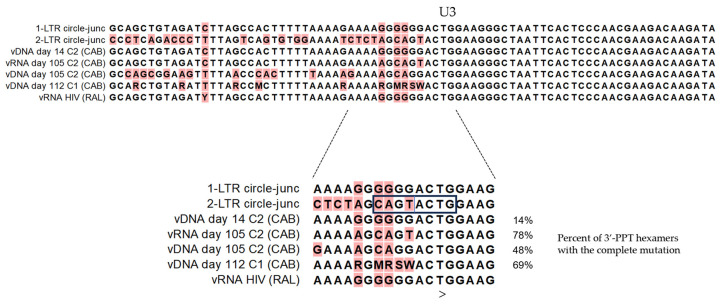
Alignment of the sequence around the 1-LTR as compared to the 2-LTR junction involved in IN recognition and cleavage and to the 3′-PPT of virus under the different treatments. Lower magnified alignment: The GTAC motif created by IN processing of the HIV termini is centered in an octamer palindrome known to be targeted for high-level endonucleolytic cleavage by integrase. Cleavage position is indicated by the central vertical line (|). The mutated 3′-PPT sequence of the vRNA and a minor sequence of vDNA at day 105 on CAB replicates the 2-LTR circle cleavage sequence. C1 = culture 1, C2 = culture 2. The ambiguities in C1 are a result of a two-nucleotide frameshift relative to the 2-LTR junction sequence. No 3′-PPT mutations were observed under RAL treatment. >, indicates the beginning of the LTR U3.

**Figure 4 viruses-16-01874-f004:**
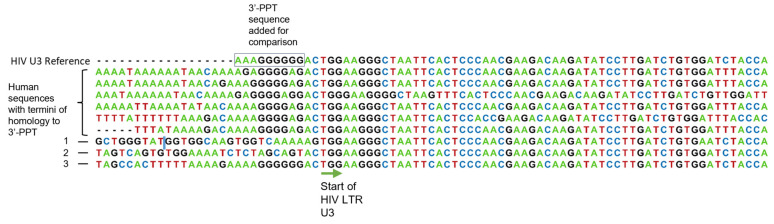
Alignment of different integrations with human DNA. The top seven sequences are human ALU to HIV gag amplification reactions showing the interface with HIV LTR. The top six sequences end in human regions homologous to the 3′-PPT. Sequence “1” shows a 20 bp fragment of the beginning of nef juxtaposed to the 3′ of human DNA where indicated by the blue line (|); Sequences “2” and “3” are HIV nef to gag 2-LTR circle amplifications showing the interface with LTR; however, for sequence 2 there is human DNA inserted ~500 bp upstream at the beginning of nef, and for sequence 3 a longer fragment of 3′-PPT and adjacent nef follows the insertion of human sequence. Complete sequences are available at GenBank (submission ID BankIt2888152).

**Figure 5 viruses-16-01874-f005:**
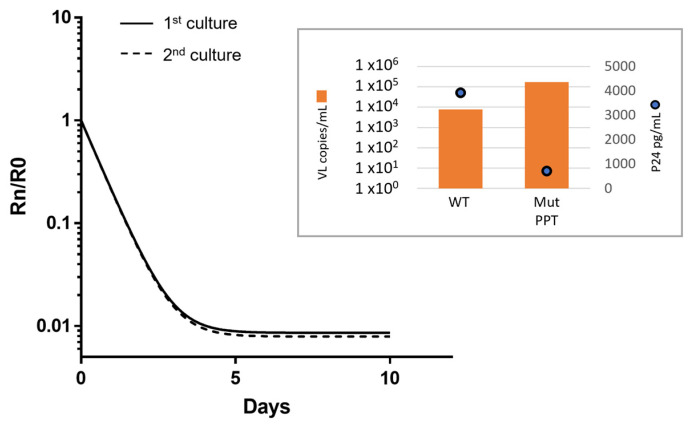
Decline of 3′-PPT HIV variant when competed with wildtype virus in the absence of CAB. The experiment performed in duplicate illustrates a rapid decline of mutant virus when co-cultured with wildtype (WT) virus. Rn/R0, proportion of mutant versus wildtype virus in culture, relative to the proportion at day 0, which plateaus to <1% at day 5. Inset: The comparison of baseline VL and p24 detected in WT virus versus the virus with the complete 3′-PPT mutation (Mut PPT).

**Table 1 viruses-16-01874-t001:** RT-PCR oligonucleotides and positions relative to the HIVIIIB genome (#-Pair number).

(RT)-PCR Primer Pair	Genomic Region	Oligonucleotide Sequence 5′-3′	Nt Positions in HIV-1IIIB Genomic DNA
1	int	F: TCTAGCTTTGCAGGATTCGG R: CAGTCTCTTTCTCCTGTATG	4034–4053 5293–5312
2	Nef-3′LTR R	F: CTAACGCTGATTGTGCCTGGC R: TGTACAGCCAAAAAGCAGCTGC	8981–9001 9550–9570
3a	2-LTR cross junction (short)	F: ATCTGAGCCTGGGAGCTCTC R: GTACTAGCTTGTAGCACCATC	9593–9612 130–150
3b	2-LTR cross junction (long)	F:TCAAGTAGTGTGTGCCCGTCTGT R: GCAGCTGCTTATATGTAGGATCTGAG	9667–9689 414–439
4	1 and 2 LTR circle	F:CTAACGCTGATTGTGCCTGGC R:TAATAYTGACGCTCTCGCACCCAT	8981–9001 790–813
5	HuAluF HuAluR To gag	CTCCCAAAGTGCTGGGATTACA TGTAATCCCAGCACTTTGGGAG TAATAYTGACGCTCTCGCACCCAT	N/A N/A 790–813
6	HuAluF HuAluR To nef	CTCCCAAAGTGCTGGGATTACA TGTAATCCCAGCACTTTGGGAG CTAACGCTGATTGTGCCTGGC	N/A N/A 8981–9001
7	Nef-2-LTR junction	F:CTAACGCTGATTGTGCCTGGC R:GAGTGAATTAGCACTTCCAGTAC	8981–9001 Junction (−2)–19
8	Viral load	F:TGCTTAAGCCTCAATAAAGCTTGCCTTGA R:TCTGAGGGATCTCTAGTTACCAG	515–543 581–603

**Table 2 viruses-16-01874-t002:** Percentage of 3′-PPT hexamer sequences containing any mutations in the AGCAGT variant (A) or GGAGCA variant (B) in the RT-PCR products examined.

**(A) AGCAGT variant**						
	**Culture 1**			**Culture 2**		
**Day**	**Nef-U3 (#2)**	**Nef-gag (#3a)**	**Nef-junct * (#7)**	**Nef-U3 (#2)**	**Nef-gag (#3a)**	**Nef-junct * (#7)**
14	NS	NS	NS	8	<2%	NS
30	NS	NS	NS	26.6	<2%	NS
98	10.1	34.2	13.9	62.3	75.6	52.9
112	5.9	13.3	1.7	78	66	55.7
120	3.1	11.2	3.3	53.1	50.9	35.6
127	3.5	9.3	1.7	42.7	51.4	24.2
**(B) GGAGCA variant**						
	**Culture 1**					
**Day**	**Nef-U3 (#2)**		**Nef-gag (#3a)**		**Nef-junct * (#7)**	
98	78.2		51.9		13	
112	90		81.6		17	
120	93.7		84.9		27	
127	93.8		82.6		16.5	

* 2-LTR-specific products; NS, not sequenced; (#), the reaction number indicated in Table 1.

## Data Availability

The sequence data will be available at GenBank.

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
