# Peer review of "HIV Replication Under High-Level Cabotegravir Is Associated with the Appearance of 3′-PPT Mutations, Circular DNA Transcription and Recombination"

_viruses, 2024, doi:10.3390/v16121874_

Round 1
Reviewer 1 Report
Comments and Suggestions for Authors
The paper by Wei et al. conducted an in vitro selection experiment to study HIV Cabotegravir (CAB)-resistant mutations. They found a small portion of the virus bearing mutations in the 3’-PPT region in the presence of high concentrations of CAB after a month of selection and in the majority of the virus by day 98. They further showed that the 3’-PPT mutations correlated with the accumulation of HIV 1-LTR and 2-LTR products. Additionally, they found an association between CAB treatment and 2-LTR recombination with human DNA. CAB has been shown to provide a higher genetic barrier and greater potency than its analogue, dolutegravir (DTG). Most importantly, CAB has been approved by the FDA as a long-acting antiviral regimen available via oral and injectable administration. To my knowledge, this is the first study to examine resistance mutations outside the integrase region under CAB treatment. Although this is an in vitro system, and several studies have been focused on DTG, however understanding CAB-related resistance mutations outside the integrase region will provide important information for clinical practice as the number of patients using CAB increases. However, several issues regarding the results and writing need to be further addressed.
1. The results are difficult to follow because the authors describe percentages and numbers in the text rather than using graphics or tables. It would benefit the reader if the authors could provide a table comparing the percentages of related products they found in the presence of CAB (3’-PPT mutations, 2-LTR, 1-LTR, LTR recombined with human DNA) for both Culture 1 and 2.
2. The study shows that 1-LTR is the major product associated with the outcome of 3’-PPT mutations. Instead of focusing on human DNA in 2-LTR, is there any evidence of human DNA recombination with 1-LTR?
3. It is unclear how the authors selected the resistant mutant virus in the presence of CAB. Graphs of HIV replication kinetics in the presence of CAB should be included in the results section.
Reviewer 2 Report
Comments and Suggestions for Authors
Overall, this is an intriguing paper. However, the results are presented in a much confusing way that prevents interpreting –and at time understanding—the results. The authors must present full results in a much clearer way before publication.
The authors must thoroughly revise their manuscript with an extensive focus on the results and figures. More details are provided below.
Results presented in Figure 2.1.1 are very confusing and would benefit from an extended figure 3. The revised figure 3 should include the origin of each sample and their specificity (e.g., 2-LTR junction or all HIV RNA). This figure should also indicate clearly which sequences are the results of junctions and which part is 3’ vs. 5’ LTR sequences (e.g., with colour coding). RNA sequencing results must not be aligned with the 2-LTR junction since this junction is only produced in the form of DNA products. Please provide a figure that fully represent the sequencing results of all species in all conditions.
An alignment (longer than 12 nucleotides) of the RNA and/or DNA sequencing results comparing selection 1 vs. selection 2 must be presented.
The deletions barely mentioned in the description of the results should be fully reported too. These deletions may have profound impact on the functionality of the gene they encompass or otherwise affect reverse transcription or integration. The authors must report their full results and not limit their report to 12 nucleotides (as it is currently the case in Figure 3).
The percentages of intact and mutated sequences are also presented in a very confusing fashion. The authors must provide a table clearly stating the total number of sequences obtained, sequence with deletions upstream/downstream of either (i) the 2LTR junction in DNA or (ii) the 3’-PPT in RNA, and sequences with the mutations of interest. The way the data is currently presented (with an * in Figure 3, and almost exclusively in the text) does not easily allow the reader to understand whether the 3’-PPT mutations are minor or major contributor to the overall HIV population in culture. It is also important to clearly detail how many of these mutated sequences suffered large deletions that may imply non-viable viruses or at least profound defects in Nef function.
The authors must present an overview of the HIV replication in the culture selection experiments. Indeed, interpretation for their results would be very different if they have high levels of HIV replication vs. barely detectable replication while under CAB (as measured for example in the cell culture supernatants). Only high levels of replication would indicate CAB resistance. Anyway, these results must be reported so that a reader can fully interpret the results.
How do the authors distinguish between mRNA vs. genomic RNA? Could one possibility to facilitate the sequencing of genomic RNA in the presence of “overwhelming quantity of incomplete transcripts” line 439-440, be the use of actinomycin D or other transcription inhibitors? Please discuss and consider performing such experiment.
The designation of mutations is unclear. Is the 6-Gs stretch of the 3’-PPT the region that is mutated? The full alignment will clarify this question.
Line 127/191: “had the intact mutation” is unclear and needs clarification. I assume that the authors mean the AGCAGT mutation. Unfortunately, without context, I still don’t understand what is reported here.
Line 222: for which culture (1 or 2).
Line 222: How can HIV RNA expression waned? Is there no longer HIV replication in the culture?
Line 226: “Amplification…” it is unclear what sample was used for this analysis. Please clarify.
Line 229: “One product…” the rest of the paragraph is understandable. Conditions, samples and exact results need to be clearly explained. The authors must present the sequencing results with all information and alignments.
Paragraph starting line 250: this is also hard to follow. For example, what does “the 3’ terminus of IN” means? Please present clear alignments and edit thoroughly the manuscript.
Line 262: sequence merging is not a scientific term and it is impossible to understand what the authors are describing in most of the paragraph. Please present full results including sample origin and alignments.
Why are the integration sites only “partially” presented in Figure 4? Please present your entire data (does not mean the entire sequences if they are too long but at least present the “10+” results. Be precise, how many integration sites did you sequence? What are the full results of mapping? What do the proximal sequences look like (alignment)? In addition, the details for 3’-end sequencing are presented and 5’-integration site sequences are not reported.
Line 322: “The CAB treatment…” this statement is not understandable.
In the discussion, the authors state that their work confirm a superiority of CAB’s genetic barrier compared to RAL. However, it should be noted that RAL was used in a dose escalation study starting from sub-inhibitory concentrations, whereas CAB was used at “high dose”. Thus, this study does not represent a direct comparison of the two drugs and the results do not support the conclusion advanced by the authors. Please remove.
In the discussion, the mutated sequence reported as GGAGCA does not resemble the one reported in the DOMONO clinical trial (Lines 334-336). The DOMONO patient had GGGAGC where most of the 6-Gs stretch was conserved. This confusion can be explained by the lack of clarity of this report.
Discussion Line 339: the second part of this statement (i.e., high levels of replication of the 3’-PPT mutated virus under CAB pressure has not been shown in the paper and should be removed.
The Nef mutations described line 344- must be reported as per nomenclature (for example, M184V in RT).
Line 348: the phenotype described here is not of resistance but of reversion. Please edit.
The authors try to explain their observations in the discussion (lines 350-), but it is impossible to understand. They must provide a schematic graph similar to the one presented by Dekker et al. Not everybody is a specialized molecular virologist. Please be kind to your readers and be as didactic as possible.
The discussion should not be limited to a repetition of the results. For example, the authors should discuss how human sequences may end up within HIV’s RNA. How is that possible? Is that because of the use of immortalized cells that are defective in DNA repair mechanisms or DNA damage detection or control?
Because the results are difficult to follow, the discussion is also challenging to understand. Of note, it does not help that the authors use non-scientific terms, for example line 366 “downstream”; one would assume that this means in 5’ but unfortunately the nature of the molecule described (RNA or DNA) is not reported. Line 370: “sequence is replicated”: “duplicated”? Line 403: “in vitro”?
Did the authors considered the contribution of auto-integrant DNA circles to the formation of some of the recombinants that they observed? Could they discuss this possibility?
In the discussion: could the various mutations in 2LTR junctions between conditions 1 vs. 2 be related to half-site integration rather than concerted integration? To answer this question, the display of both integration sites would be helpful.
Statement starting line 416 is unclear, please edit.
Line 420 change “to” to “than”
Sentence starting line 419 is unclear, please edit.
Line 57: I don’t know that CAB is a “more stable analog of DTG”. Please provide a reference or edit. The authors may be confusing PK data with intrinsic stability.
In the Methods, line 67: the reagents were obtained from the NIH AIDS Reagent Program and should be acknowledge appropriately. BEI Resources is a management platform.
Line 92: “Later experiments”? Can the authors specify which samples were analyzed with which method?
Line 207: “the vast majority” is too vague, please provide an exact number.
Results presented on line 248 are intriguing, can the authors discuss in the discussion what advantage may be derived from a duplicated integrase processing site with this HIV sequence?
Round 2
Reviewer 1 Report
Comments and Suggestions for Authors
The author has addressed my questions.
Reviewer 2 Report
Comments and Suggestions for Authors
Thank your efforts to improve the clarity of your manuscript. I believe that your paper is now ready for publication.